# The Influence of Surface Treatment of PVD Coating on Its Quality and Wear Resistant

**Tomas Zlamal \*, Ivan Mrkvica, Tomas Szotkowski and Sarka Malotova**

Faculty of Mechanical Engineering, VŠB – Technical University of Ostrava, 17. listopadu 2172/15, 708 00 Ostrava-Poruba, Czech Republic

\* Correspondence: tomas.zlamal@vsb.cz; Tel.: +420-597-324-329

**Abstract:** The article deals with a determination of the influence of a cutting edge preparation on the quality and wear resistance of coated cutting tools. Cutting inserts made from a sintered carbide with a deposited layer of PVD coating were selected for measurement. Non-homogeneity caused by the creation of droplets arises in the application layer during the process of applying the coating by the PVD method. These droplets make the surface roughness of the PVD coating worse, increase the friction and thereby the thermal load of the cutting tool as well. Also, the droplets could be the cause of the creation and propagation of droplets in the coating and they can cause quick cutting tool wear during machining. Cutting edge preparations were suggested for the improvement of the surface integrity of deposited layers of PVD coating, namely the technology of drag finishing and abrasive jet machining. After their application, the areal surface roughness was measured on the surface of coated cutting inserts, the occurrence of droplets was tracked and the surface structure was explored. A tool-life test of cutting inserts was carried out for verification of the influence of surface treatment on the wear resistance of cutting inserts during the milling process. The cutting inserts with a layer of PVD coatings termed as samples A, B, and C were used for the tool-life test. The first sample, A, represented the coating before the application of cutting edge preparations and samples B and C were after the application of the cutting edge preparation. A carbon steel termed C45 was used for the milling process and cutting conditions were suggested. The visual control of surface of cutting inserts, intensity of wear and occurrence of thermal cracks in deposited PVD layers were the criterion for the evaluation of the individual tests.

**Keywords:** PVD coating; cutting edge preparation; droplets; surface roughness; wear

## 1. Introduction

Increasing attention is being paid to cutting edge preparation and its surface integrity [1–3]. It is scientifically justified that its state directly influences the wear and tool life of the cutting tool [4–6] and the intensity of the thermal and mechanical loads [7]. Kim, et al. [8] concluded that an increased cutting edge radius causes change in the temperature distribution of the cutting tool. Next, adhesion of the applied coatings and their sliding characteristics, chip forming, and the quality of the machined surface have been studied in publications [9–11]. This was, for example, the subject of the solution by authors Zhao et al., who mentioned the influence of cutting tool microgeometry on the machining process [12].

Microgeometry is defined by shapes occurring on the cutting edge, surface roughness, as well as by microscopic defects emerging on the surface during pressing, sintering, and peripheral and frontal grinding. Cutting edge preparation is put into the manufacturing process for the removal of defects, increasing the total performance and reliability of the cutting tool. [13] This technology of cutting edge preparation is most often associated with edge rounding for cutting inserts made of sintered

carbide. Denkena et al. also summarized the individual methods of preparation and their influence on the surface integrity of cutting tools in publications [13–16]. The author Rodriguéz [17] published book describing the cutting edge preparation of precision cutting tools, current state, methods and evaluations focused on this issue.

Cutting edge preparation is integrated into production mainly due to the very low mechanical strength of the cutting edge and an inclination to pull out the hard carbides from the binder during contact with the material being machined. Except for the removal of defects in the vicinity of the edge and the rounded edge, which are defined by their dimensions and shape with high mechanical strength, arise by means of cutting edge preparation technology. Microtopographical changes of the cutting edge and microstructural changes on the rake face and flank face of the cutting tool are an inseparable part of this process. In particular, these changes are important from the point of view of the coating process because they influence adhesion between the substrate and the applied coatings that accurately copy the surface structure of the cutting tool [18,19]. The influence of cutting edge preparation on adhesion of applied PVD and CVD coatings are described by authors Lukaszokowicz, et al. [20]. Work by Knotek et al. [21] concludes that coating adhesion increases with improved surface quality and low surface roughness values offer more favorable growth conditions for hard PVD coatings. This increases micro-hardness and the coating rate.

The cutting edge preparation process is also important with respect to decreasing the residual stress that was brought into the substrate during production and grinding afterwards. The occurrence and high concentration of stress in the undersurface and surface layers are very often the initiators of the formation and propagation of cracks, as well as adhesive or cohesive damage of the coatings and edge chipping [22]. The authors Breidenstein and Denkena et al. described the significance of residual stress in sintered carbide cutting tools in their publication [23], and in their other research [24,25] they deal with residual stress in applied coatings layers. The authors Oettel and Wiedemann evaluated residual stress in PVD coatings. Their distribution can have an effect on a behavior of coated materials that is caused by the thermoelastic effects and grown-in defects [26].

The formation of residual stress occurs during the coating of the substrate as well. However, depending on the type of applied coatings and the appropriate cutting edge preparation after coatings, the residual stress can be eased or can change the character of their effect. Residual stress can be changed from tensile to compressive in the applied coatings layers of coatings at suitable selected cutting edge preparations. This change is suitable especially for cutting tools which are exposed to intensive changes of temperature. The presence of compressive stress in the layers of PVD coatings can influence the creation and spreading of thermal cracks [27]. The authors Bhatia et al., in their publications [28,29], dealt with thermal cracks and their influence on the wear process of cutting tools during intermittent cutting processes. The mechanical shocks cause edge chipping of cutting inserts and thermal cracking is correlated by temperature distribution and thermal stress at the tool face during the cutting process. Droplets and other unevenness on the surface of the cutting tool that the coatings copied before adjustment can be removed by cutting edge preparation. This results in lower surface roughness and better sliding characteristics in the deposited layer. Knowledge gained about the decrease in friction and also in the thermal load of the cutting edge as a result of lower surface roughness is mentioned in [30,31]. Coated tools compared with uncoated ones offered better protection against mechanical and thermal loads, diminish friction and the interaction between tool and chip, and improve wear resistance in a wide cutting temperature range [32].

## 2. Methods of the Cutting Edge Preparations

Cutting edge preparations are made in particular to remove defects and surface unevenness occurring on the cutting edge and its surroundings, to round the edge and to improve the surface integrity of the cutting tool before and mainly after the coating process as well. The size and shape of the edge radius, surface structure, surface roughness and required functional characteristics of the coatings are determined primarily by the appropriate choice of technology.

Currently, several technologies of cutting edge preparation are used, and their sorting and application have been clearly described in the technical articles [14,17,33]. Despite this, development in this area is focused on searching for new technologies, but also on innovation of the technologies that are used daily, in serial and mass production. The aim of the search is to achieve within the largest possible control of process parameters, high repeatability, productivity and automation.

Technologies of cutting edge preparation such as abrasive jet machining and drag finishing were proposed for the purpose of experimental activity. These technologies are based on the principle of the mechanical interaction of abrasive particles. These technologies can be used for cutting edge preparation of cutting tools before coating, and in some cases, after the coating process. The amount and intensity of material taken is dependent on the kind of abrasive used, the designed process parameters, and the length of their interaction [17].

### 2.1. Abrasive Jet Machining

Abrasive jet machining removes defects from surfaces with a strong jet of abrasive particles and water. An advantage of using this technology is its low dustiness, as well as the small amount of settled medium residues compared with other surface technologies. Better surface roughness and lower residual stress in the substrate, or applied coatings, is achieved in cutting tools using abrasive jet machining. A decrease in the number of droplets from the surface of the coating and improvement of tribological characteristics is expected during treatment of the applied PVD coating with abrasive jet machining. The impact of high kinetic energy abrasive particles on the coating's surface also improves its strengthening and crack propagation resistance in the coating [17,34]. Parameters of abrasive jet machining (Figure 1) were used for the surface treatment of coated cutting inserts used for the experimental activity, see Table 1.

**Table 1.** Parameters of abrasive jet machining.

| Kind of Medium | Blasting Distance | Belt Feed Rate | Blasting Angle | Air Pressure | Concentration | Process Time |
|---|---|---|---|---|---|---|
| pink $Al_2O_3$ - 240/280 mesh | 170 mm | 65 mm/min | ±30° | 2.5 bar | 80% water 20% medium | 6 min/100 pcs |

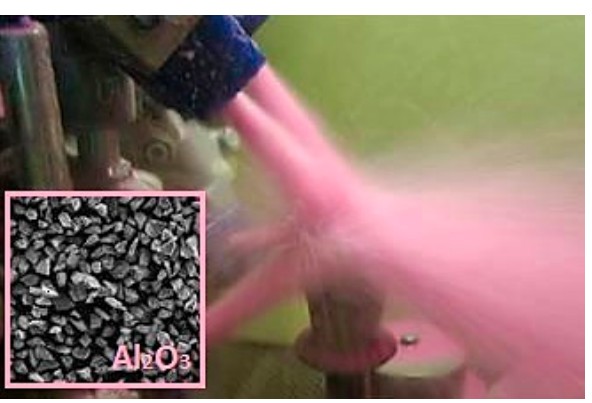

**Figure 1.** Abrasive jet machining process.

### 2.2. Drag Finishing

As in abrasive jet machining, drag finishing (see Figure 2) belongs within the technology of mechanical surface treatments. Material is removed due to the abrasive effect of mutual movements between the surface of the solid figure and the abrasive medium during drag finishing [12]. Different mechanical dirt is removed from the surface of the solid figure, depending on the drag finishing technology used, the kind of medium, the parameters, and the drag finishing time. Drag finishing technology is most commonly used for grinding, surface polishing, as well as sintered carbide cutting

edge preparation. In this case, it is possible to use drag finishing technology before and after the coatings process. Cutting tools are dragged in a planetary motion container at high speed during drag finishing technology. This way, the desired pressure is achieved between the cutting tool surface and the process medium. Surface treatment of cutting inserts took place during the dry process of drag finishing with a walnut granulate. A combination of grinding paste is particularly suitable for fine edge rounding for a very smooth and glossy surface or for cutting tools made of sintered carbide or cutting ceramics [35]. The following process parameters for drag finishing were used, see Table 2.

**Table 2.** Parameters of drag finishing.

| Main Rotor | Small Rotor | Depth of Draft | Process Medium | Grain Size |
|---|---|---|---|---|
| 30 rpm | 50 rpm | 95 mm | walnut shell granulate impregnated with polishing paste | 0.8–1.3 mm |

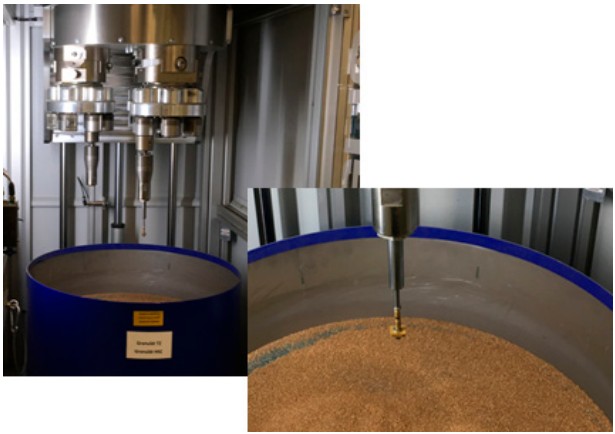

**Figure 2.** Drag finishing.

## 3. Measurement and Evaluation of Surface Quality of Cutting Inserts

Verification of the mechanical effects of surface treatments on the achieved surface quality was carried out on selected exchangeable cutting inserts made of sintered carbide with an applied layer of PVD coatings (see Table 3). The assessment was based on the measuring of surface microgeometry, the determination of the occurrence of droplets, and the evaluation of surface structure of PVD coatings before and after cutting edge preparation. The samples of cutting inserts without surface treatments were marked with the letter A, after abrasive jet machining with the letter B, and after drag finishing with the letter C.

**Table 3.** Coating and substrate of used cutting inserts.

| Type of Insert: SPKN 1203EDER-M | | |
|---|---|---|
| **Coatings** | **PVD coatings - thickness 4 μm** | |
| | TiN + multiplelayer structure TiAlN/TiN + TiN | |
| | (approx.: TiN—0.4 μm; TiAlN/TiN—3.4 μm; TiAlN—0.2 μm) | |
| **Substrate** | sintered carbide | |
| | W = 87.1% | |
| | Cr = 0.7% | |
| | C = 3% | |
| | Co = 92 % | |

Measurement of the surface microgeometry of coated cutting inserts was carried out using an Alicona InfiniteFocus G5 optical microscope (Graz, Austria) and MeasureSuite software (version 5_3_4) Using the microscope, high-quality surface detail in high resolution was obtained very quickly and accurately. By adjusting the resolution, it was possible to identify different kinds of defects and render the surface in real colors. Since the surface of the cutting inserts is aperiodic after the coating process and technologies of cutting edge preparation, selected areal surface parameters were used for evaluation. They set with more accuracy information about the surface structure and its quality. In contrast to the profile surface roughness, the whole area of the specimen will be evaluated. This measurement was realized on rake face $A\gamma$ at a distance of approximately 1.5 mm from the tool tip. Values of areal surface parameters are shown in Table 4.

**Table 4.** Microgeometry measurement.

| Parameter | Microgeometry Measurement – Magnification 50x | | |
|---|---|---|---|
| | Vertical Resolution [μm] | Lateral Resolution [μm] | Cut-Off Filter $\lambda_c$ [μm] |
| | 0.10 | 0.44 | 60.80 |
| | Specimen – A | Specimen – B | Specimen – C |
| $Sa$ [μm] | 0.21 | 0.11 | 0.14 |
| $Sz$ [μm] | 4.83 | 3.02 | 4.19 |
| True color information | | | |
| Pseudo color information | | | |

The influence of cutting edge preparation on the surface of the PVD coatings was evident from the measured values of parameters. The cutting edge preparation caused smoothing, and thus a decrease of the surface roughness value against the unprocessed surface. The resulting quality was also influenced by the quality of the foundation surface and the amount of excreted droplets after the coating process with the coating deposited by PVD methods. However, the removal of droplets by cutting edge preparation does not necessarily improve surface roughness. The droplets are embedded in the coating and their pulling out causes dimples on the surface. The resulting surface roughness can differ only slightly after the PVD coating. The number of excreted droplets before and after the cutting edge preparation of coated cutting inserts was determined using a MIRA3 LM scanning microscope, and multiple magnified details were created (see Table 5).

**Table 5.** Occurrence of droplets before and after cutting edge preparation.

| Specimen A – Coating without Cutting Edge Preparation | |
| :---: | :---: |
| Magnification – 2000x | Magnification – 5000x |

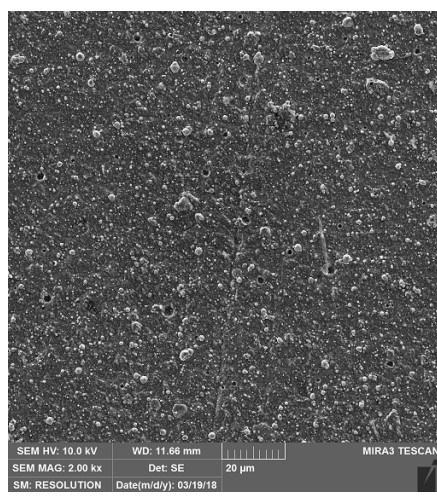 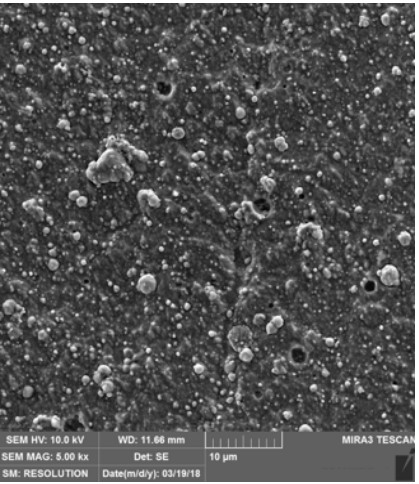

| Specimen B – Coating after Abrasive Jet Machining | |
| :---: | :---: |
| Magnification – 2000x | Magnification – 5000x |

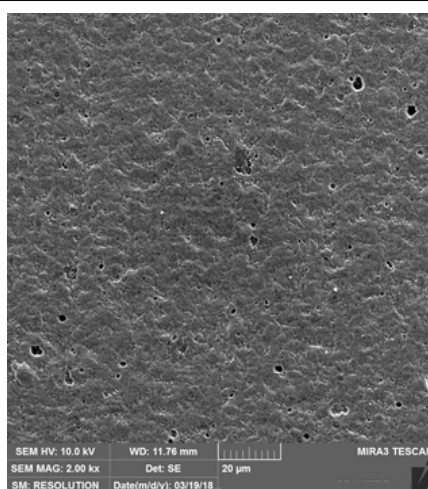 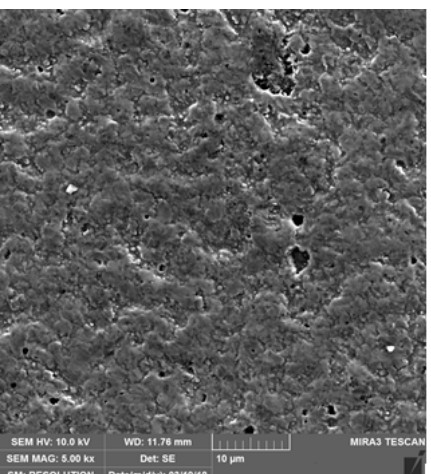

| Specimen C – Coating with Drag Finishing | |
| :---: | :---: |
| magnification – 2000x | magnification – 5000x |

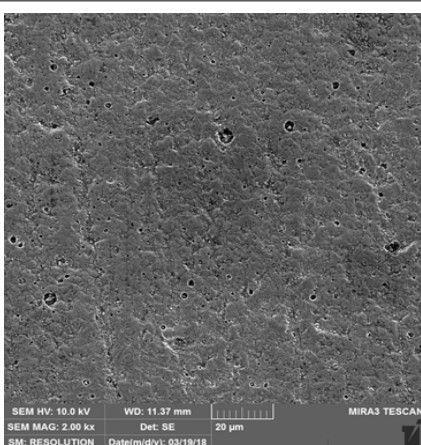 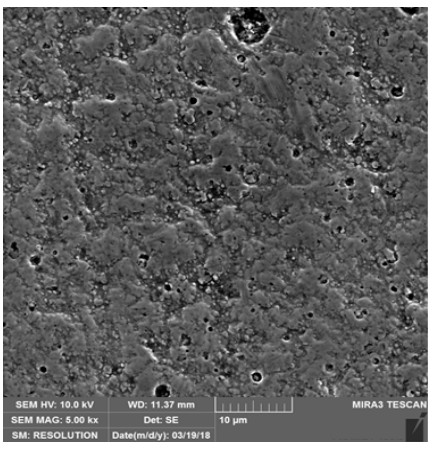

The number of excreted droplets can be observed from the figures before and after the cutting edge preparation of the PVD coating. The droplets make the surface roughness worse and increase the friction on the cutting edge in the unprocessed specimen. They were removed by the cutting edge preparation technology, but at the same time, hollows were created on the surface. However, unlike the droplets, they did not have a negative effect during the cutting edge load, and so there is no intensive friction because of the cutting liquid adhering to the surface. In the figures, it is possible to observe visible traces caused by the previous treatment of the substrate that the PVD coating copied.

The smoothest surface was achieved with drag finishing, which was confirmed by the measurement of the areal surface roughness. The results may be distorted by dirt contained in the hollows after the droplets, and in the traces of the previous treatment, see Figure 3. Surface pollution does not occur using abrasive jet machining because of the high impact speed of the abrasive medium on the surface.

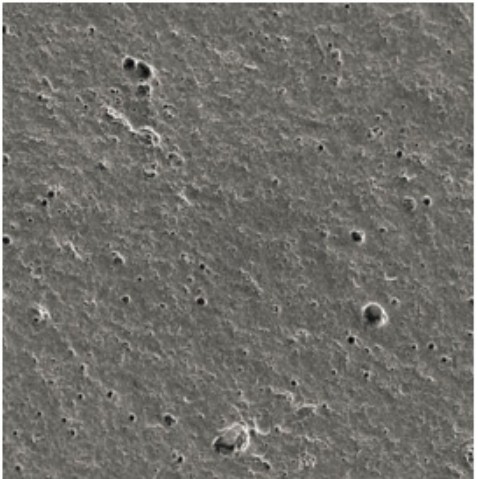 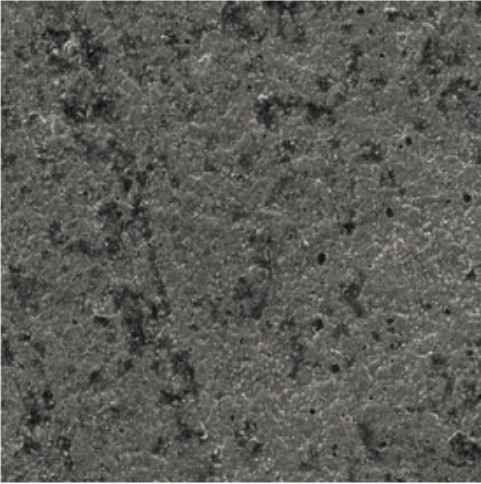

**Figure 3.** Image of dirties on the surface of tested PVD coating after cutting edge preparation (dark areas) **Left**—abrasive jet machining, **Right**—Drag finishing.

A tool-life test for the milling process was designed for the determination of the PVD coating resistance against wear. The milling tests were performed in the form of face milling, with the tool and under milling conditions as described in Table 6. The cutting conditions were chosen on the basis of recommendation and realized functional tests. The chosen machined material was a medium carbon non-alloyed steel C45 according to DIN. The coolant liquid was directed straight to the outcoming insert from the cut in order to produce as much thermal shock as possible on the edge of the investigated inserts. The milling tool-life test was realized for 24 pieces of cutting inserts.

**Table 6.** Face milling conditions and parameters.

| | | |
|---|---|---|
| **Mill diameter** | $D = 100$ mm | 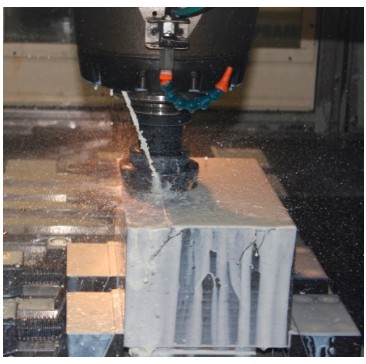 |
| **Entering angle** | $k_r = 75°$ | |
| **Cutting speed** | $v_c = 400$ m/min | |
| **Revolution** | $rev. = 1256$ rpm | |
| **Cutting depth** | $a_p = 2.5$ mm | |
| **Feed per tooth** | $f_z = 0.2$ mm | |
| **Cutting width** | $a_e = 50$ mm | |
| **Cutting length** | $L = 330$ mm | |
| **Cooling** | YES | |

The resistance of the PVD coating against wear was evaluated on the basis of the origins of cracks during the milling process. Their presence was found by a visual inspection of multiple magnified images of the cutting edge using the Keyence VHX 6000 optical microscope (Mechelen, Belgium). Crack size, number of cracks, and their impact on cutting edge quality was evaluated. Due to the growth and character of crack propagation in the coating and substrate, it is also necessary to focus on the distance between the cracks. A small distance used to be the cause of early breakage and destruction of the cutting edge. In Table 7, it is possible to observe the amount and size of cracks together with edge chipping of cutting tool.

**Table 7.** Thermal cracks after milling process.

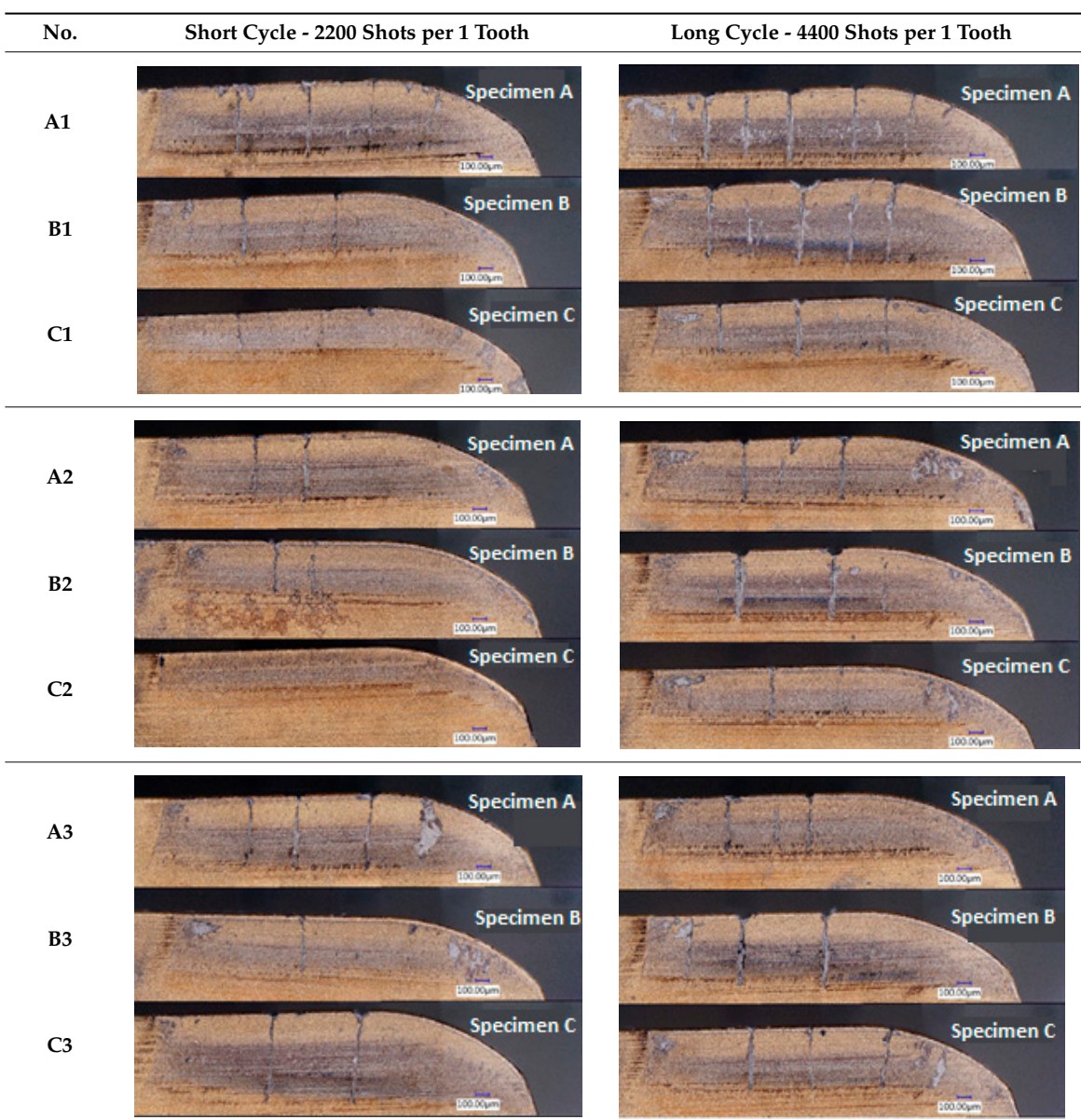

| No. | Short Cycle - 2200 Shots per 1 Tooth | Long Cycle - 4400 Shots per 1 Tooth |
|-----|--------------------------------------|-------------------------------------|
| A1 | | |
| B1 | | |
| C1 | | |
| A2 | | |
| B2 | | |
| C2 | | |
| A3 | | |
| B3 | | |
| C3 | | |

On the basis of visual evaluation, the variant C—cutting edge preparation of drag finishing, was the best in terms of the creation of cracks, their amount and edge chipping. Faster creation of thermal cracks was observed in variant A—unprocessed surface, and variant B—abrasive jet machining. Their propagation caused visible edge chipping in the cutting scene. Very similar results were achieved for the other 24 pieces of cutting inserts tested.

## 4. Discussion of the Results

The aim and intent of the research was to design an appropriate cutting edge preparation for the achievement of higher surface quality and better thermal cracks resistance. Milling cutting inserts marked SPKN 1203EDER-M with PVD coating based on TiAlN + TiN were selected for milling process. The droplets were created and excreted on a surface of PVD coating during the process of depositing the PVD coating, see Figure 3. Their occurrences and amount deterioration of the coating surface roughness and its sliding characteristic, therefore the friction and thermal load of the cutting edge depend on that.

Cutting edge preparation, drag finishing and abrasive jet machining were designed for droplet removal from the surface of the PVD coating. The parameters of these technologies were selected with regards to a higher surface quality and as little possible damage or making thinner layers of the PVD coating. The parameters of surface roughness Sa and Sz were measured to determine the influence of the cutting edge preparation and designed parameters on surface quality of the PVD coating. Surface quality was also evaluated from images taken using a scanning microscope. In the images of the PVD coating surface before and after cutting edge preparation, it is possible to observe the number of excreted droplets, their removal, and the unevenness of the surface foundation that the coating copied after previous treatment, see Table 5.

Depending on the process time and the medium used, a very smooth and glossy surface was achieved in both cases on the coated cutting inserts. A tool-life milling test was performed to determine the influence of the cutting edge preparation used on wear resistance. In total, 24 pieces of cutting inserts were tested during the face milling process of material DIN C45 under the determined cutting conditions. The formation of thermal cracks, their amount and edge chipping were observed on the basis of visual inspection.

Lower values of Sa and Sz parameters were achieved during measurement of the surface roughness of the cutting inserts tested marked B—abrasive jet machining and C—drag finishing. Improved surface roughness was achieved by removing the droplets from the PVD coating and also by smoothing out the unevenness that the coating copied after previous treatment. The difference between the values was not that much considering it was caused by anchoring of droplets to applied coating and creation of the hollow after their removal. Comparison with an unprocessed surface of PVD coating could show an improvement in sliding characteristics, a decrease in friction and a smaller thermal load on the cutting. These changes should have manifested during the milling process when the tested cutting tools were exposed to very rapid temperature changes. Nevertheless, the wear started to manifest on tested cutting inserts caused by the creation of thermal cracks and their propagation, see Table 7 above. Higher crack resistance and crack propagation has been achieved with variant C—drag finishing. A lower number of cracks and edge chipping caused by thermal cracks were proved by visual inspection. At the same time, the average length and edge chipping was determined with the amount of the thermal cracks on the cutting edge, see Table 8. The principle of measuring thermal cracks is shown in Figure 4.

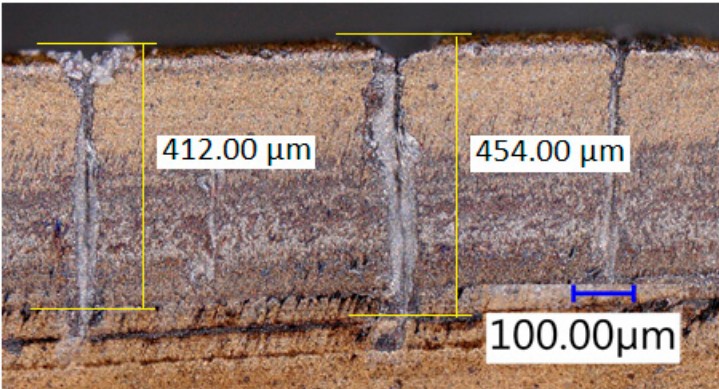

**Figure 4.** Measurement methods of the length of thermal cracks.

**Table 8.** Average length of thermal cracks.

| Period | Short Cycle - 2200 Shots per 1 Tooth | | | Long Cycle - 4400 Shots per 1 Tooth | | |
|---|---|---|---|---|---|---|
| Specimen | No. Cracks | Average Length [μm] | Chipping Yes/No | No. Cracks | Average Length [μm] | Chipping Yes/No |
| A1 | 4 | 487 | No | 5 | 496 | Yes |
| B1 | 2 | 412 | Yes | 3 | 566 | Yes |
| C1 | 2 | 295 | No | 3 | 370 | No |
| A2 | 2 | 390 | No | 2 | 409 | Yes |
| B2 | 2 | 308 | No | 2 | 512 | Yes |
| C2 | 0 | 0 | No | 2 | 247 | No |
| A3 | 3 | 398 | No | 2 | 355 | No |
| B3 | 1 | 401 | No | 3 | 470 | No |
| C3 | 2 | 607 | No | 3 | 400 | No |

Thermal cracks and cracks propagation occurred with variants A and B. There has also been a higher amount of edge chipping. Incorrect process parameter settings of the abrasive jet machining were the cause of a decrease in PVD coating resistance with variant B. The intense effect from the very hard grains of the abrasive medium ($Al_2O_3$) resulted in a reduction of thickness or damage of the PVD coating integrity. This was also the case with a variant C. The surface of the coatings was damaged due to incorrectly selected process parameters of drag finishing. It was the cause of the formation and propagation of a larger number of thermal cracks during the milling process, see Figure 5.

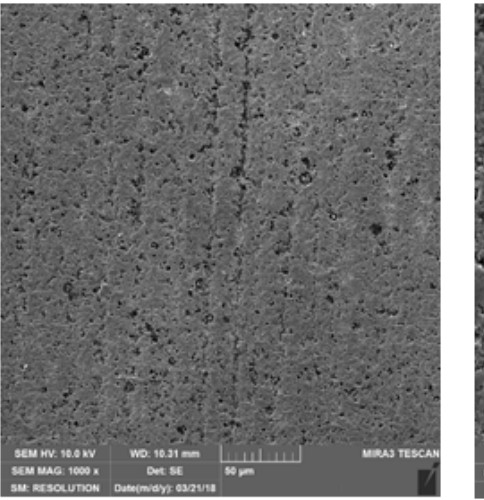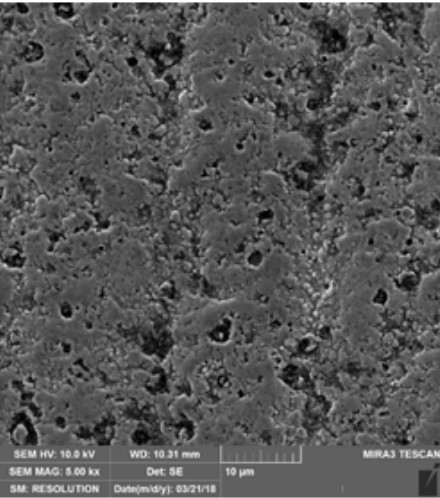

**Figure 5.** Damage of PVD coatings during drag finishing (time = 10 min). **Left**—magnification 1000x; **right**—magnification 5000x.

## 5. Conclusions

From the point of view of an evaluation of the surface quality of cutting tools, it is necessary to select the correct cutting edge preparation and its process parameters. In spite of achieving a glossy look with very low surface roughness, the designed cutting edge preparation may damage the applied coating enough to lose its functional characteristics. However, current technology and production processes are not able to manufacture a cutting tool with the required accuracy and quality. For this reason, cutting edge preparation is inevitable and it is necessary to devote attention to this research area. Their character and effect on the functional properties of the cutting tool are necessary also to experimentally verify for detailed research.

- Select cutting edge preparation and process parameters with regards to type and thickness of applied coating layer;

- Ensure higher surface quality before coatings due to copy the unevenness;
- Reduce the number of droplets and unevenness due to improvement surface roughness and sliding characteristics of the coatings;
- Ease or change the character of the effect of the residual stress in order to increase the fatigue life of applied coatings layer.

**Author Contributions:** Conceptualization, T.Z. and I.M.; methodology, T.Z. and S.M.; validation, T.Z.; I.M.; S.M. and T.S.; investigation, S.M.; resources, T.S.; data curation, S.M., T.Z. and T.S.; writing—original draft preparation, T.Z. and I.M.; writing—review and editing, S.M.; supervision, I.M.

**Funding:** This work has been done in connection with projects Education system for personal resource of development and research in field of modern trend of surface engineering - surface integrity, reg. no. CZ.1.07/2.3.00/20.0037 financed by Structural Founds of Europe Union and from the means of state budget of the Czech Republic and by the Specific Research Project SP2019/60 financed by the Ministry of Education, Youth and Sports and Faculty of Mechanical Engineering VŠB-TUO.

**Conflicts of Interest:** "The authors declare no conflict of interest." "The funders had no role in the design of the study; in the collection, analyses, or interpretation of data; in the writing of the manuscript, or in the decision to publish the results".

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
