# Peer review of "The Influence of Surface Treatment of PVD Coating on Its Quality and Wear Resistant"

_coatings, doi:10.3390/coatings9070439_

Round 1
Reviewer 1 Report
The paper presents the results of original research. Article, however, requires more detailed explanations and additions.
Comments and questions:
in section 1:
The introduction should be improved. In my opinion the literature analysis is too short (there are only 11 references) and requires more details. Especially the wear mechanism of cutting tools with PVD coatings should be described better with reference to literature. References should be done in one to one mode for better clarity.
in section 3:
- No information about the carbide grade of the cutting insert. There is also no information on the number of coatings and their type (material).
- Has the type of treatment affected the thickness of the coating layer? Have cross sections been made for the cutting insert? Does the type of machining (abrasive jet machining and drag finishing) affect the cutting edge radius values?
- line 163 - Table 4 - It is not clear what these pictures represent.
- In experimental studies, one feed rate, cutting speed and cutting depth were adopted. No explanation of the reason for this decision. Why has no experimental tests been carried out on the wear resistance of cutting inserts for variable cutting data values? No experimental research plan and results analysis. No mathematical dependencies and graphs depicting these dependencies. The article should be extended with additional test results.
- What other authors think about the creation of thermal cracks in milling with the supply of coolant?
- Why wasn’t an analysis of the wear of the cutting insert on the rake and flank face? In the article, there are no results of measurements of parameters describing the wear of cutting insert, eg VB or KT. The authors didn’t present the wear curve of the cutting insert for the analyzed case. The article should be completed.
in section “Conclusions”:
- Conclusions should be based on concrete test results. The authors only give general conclusions based on observations. For example: line 225 – “lower number of cracks and chipping…”; line 226 – “There has also been a higher amount of edge chipping”; line 228 – “… cause decrease PVD coating resistance”. What were the numerical values?
- I suggest to indicate practical conclusions resulting from the tests, especially in relation to the milling process.
Author Response
Reviewer 1 comments:
The paper presents the results of original research. Article, however, requires more detailed explanations and additions.
Comments and questions:
in section 1:
The introduction should be improved. In my opinion the literature analysis is too short (there are only 11 references) and requires more details. Especially the wear mechanism of cutting tools with PVD coatings should be described better with reference to literature. References should be done in one to one mode for better clarity.
It has been changed. The literature has been extended.
in section 3:
- No information about the carbide grade of the cutting insert. There is also no information on the number of coatings and their type (material).
An information about grade has been added.
- Has the type of treatment affected the thickness of the coating layer? Have cross sections been made for the cutting insert? Does the type of machining (abrasive jet machining and drag finishing) affect the cutting edge radius values?
The thinning of coted layer was dependent on preparation parameters. In our case, we tried to removal only just the droplets from surface without any other damages. We tested several parameters combination.
- line 163 - Table 4 - It is not clear what these pictures represent.
It has been changed.
- In experimental studies, one feed rate, cutting speed and cutting depth were adopted. No explanation of the reason for this decision. Why has no experimental tests been carried out on the wear resistance of cutting inserts for variable cutting data values? No experimental research plan and results analysis. No mathematical dependencies and graphs depicting these dependencies. The article should be extended with additional test results.
The used cutting parameters were selected on basis of the functional tests.
- What other authors think about the creation of thermal cracks in milling with the supply of coolant?
The current state has been added.
- Why wasn’t an analysis of the wear of the cutting insert on the rake and flank face? In the article, there are no results of measurements of parameters describing the wear of cutting insert, eg VB or KT. The authors didn’t present the wear curve of the cutting insert for the analyzed case. The article should be completed.
The object of our research was a presence of thermal cracks after milling process. The other wear patterns were not detected.
in section “Conclusions”:
- Conclusions should be based on concrete test results. The authors only give general conclusions based on observations. For example: line 225 – “lower number of cracks and chipping…”; line 226 – “There has also been a higher amount of edge chipping”; line 228 – “… cause decrease PVD coating resistance”. What were the numerical values?
We created the chapter “Discussion of results” and conclusion and recommendations were added.
- I suggest to indicate practical conclusions resulting from the tests, especially in relation to the milling process.
It could be considered.
Thank you for your comments.
Reviewer 2 Report
The paper “the Influence of Surface Treatment of PVD Coating on Its Quality and Wear Resistant” present a hot topic for manufacturing sector.
Despite interesting topic some comments may help to improve this paper.
I suggest to revise the format of this paper because in the actual format is quite difficult to follow it.
The section “2. Suggestion of cutting edge preparation after coatings” should be renamed as materials and methods.
A section with results and discussion is necessary!
The conclusions are to long, may be some part from conclusion needs to go in results and discussions. In the conclusions is needed a concise detail of the paper outcomes.
The scale and details of voltage is missed in Figure 4 and almost interfere with Table 4 that is difficult to read
From Abstract and introduction is not undesirable your contribution
The state of art is very brief and requires improvements
The English is very poor, it requires critical adjustments. Couples of examples that make almost impossible to understand the manning of the text as follow: “As in abrasive jet machining drag finishing (Figure 2) belongs between cutting edge preparation technologies…..”Verification of mechanical effects of surface treatments on achieved surface quality was taken place for selected exchangeable cutting inserts made of sintered carbide (Figure 3) with the applied layer of PVD coatings.”
Poor formulation such as “cutting tool also. Also, the droplets”
In table 5 you have a table and a Figure attached ? why ?
What is the difference between the two pictures in “Figure 5. Milling process under cooling liquid stream” and why you do not have caption (a) and (b)
Is quite difficult to understand from “In table 6 for coated 185 cutting inserts” which parts are coated and not..
In summary I suggest critical reviewers and update the manuscript for a technical review because in the present form this manuscript do not look as a research paper
Author Response
Reviewer 2 comments:
The paper “the Influence of Surface Treatment of PVD Coating on Its Quality and Wear Resistant” present a hot topic for manufacturing sector.
Despite interesting topic some comments may help to improve this paper.
I suggest to revise the format of this paper because in the actual format is quite difficult to follow it.
We have tried to change. The actual format is according to journal template.
The section “2. Suggestion of cutting edge preparation after coatings” should be renamed as materials and methods.
It has been changed - Methods of the cutting edge preparations.
A section with results and discussion is necessary!
It has been added.
The conclusions are to long, may be some part from conclusion needs to go in results and discussions. In the conclusions is needed a concise detail of the paper outcomes.
It has been changed.
The scale and details of voltage is missed in Figure 4 and almost interfere with Table 4 that is difficult to read
It has been added and changed.
From Abstract and introduction is not undesirable your contribution
It could be considered
The state of art is very brief and requires improvements
It has been improved.
The English is very poor, it requires critical adjustments. Couples of examples that make almost impossible to understand the manning of the text as follow: “As in abrasive jet machining drag finishing (Figure 2) belongs between cutting edge preparation technologies…..”Verification of mechanical effects of surface treatments on achieved surface quality was taken place for selected exchangeable cutting inserts made of sintered carbide (Figure 3) with the applied layer of PVD coatings.”
English quality has been improved by native speaker.
Poor formulation such as “cutting tool also. Also, the droplets”
These terms are widely used in scientific literature.
In table 5 you have a table and a Figure attached ? why ?
It has been changed.
What is the difference between the two pictures in “Figure 5. Milling process under cooling liquid stream” and why you do not have caption (a) and (b)
It has been changed.
Is quite difficult to understand from “In table 6 for coated 185 cutting inserts” which parts are coated and not..
The explanation is shown in the paragraph below Table 6. Altogether 42 coated samples were tested.
In summary I suggest critical reviewers and update the manuscript for a technical review because in the present form this manuscript do not look as a research paper.
Thank you for your comments.
Reviewer 3 Report
The article discusses the methods of surface preparation for coating. These issues were widely discussed earlier and, in general, the authors do not report any significant new information. The work must be radically redone, including conducting detailed studies on the nature of the destruction of coatings.
The abstract should be substantially revised, taking into account that the abstract is a focused presentation of the main results and ways to achieve them. Unnecessary details (such as "samples A, B, and C were used") should be removed. The usual structure of the Abstract is “Solved problem (1-2 phrases)” - “Solution methods” - “Obtained results”.
The term "durability test" is not very well used in this context. Correct to use the term "cutting test"
Since a scientific article differs from a textbook, it makes no sense to use images that do not contain new information (for example, images of standard equipment or standard processes). From this point of view, it is worth checking out Figures 1, 2 and 5. Do these figures contain anything new and useful for a professional?
Also, I see no point in Figure 3 (especially since this information is duplicated in more useful form in Table 5.)
A review of the current state is clearly insufficient. Total examined 11 articles, of which 6 - article by one author (certainly distinguished). The language of the article is not high enough. A lot of mistakes, terminological inaccuracies, English needs significant improvement.
The result of the Introduction should be a clearly defined work goal. The goal, formulated as "the largest possible control of process parameters, a high repeatability, productivity and automation" is too general and not specific.
Table 3 and 4, Figure 6 - scale bars should be added.
What coatings for cutting tools were used? Why were they chosen?
The authors talk about "lower residual stress" - but there are no measurements of these parameters.
And what is a "Sample ABC"? Where are they described?
Table 3 should be better structured.
The article is completely raw and requires radical processing. The level of research is insufficient (better study of the properties of the surface layer after various processing methods using, in particular, cross sections is necessary, at least).
The reviewer cannot recommend this article for publication.
Author Response
Reviewer 3 comments:
The article discusses the methods of surface preparation for coating. These issues were widely discussed earlier and, in general, the authors do not report any significant new information. The work must be radically redone, including conducting detailed studies on the nature of the destruction of coatings.
Currently, new devices and methods for evaluation of cutting tool wear and surface quality are available. A new PVD coating has been used for the research purpose.
The abstract should be substantially revised, taking into account that the abstract is a focused presentation of the main results and ways to achieve them. Unnecessary details (such as "samples A, B, and C were used") should be removed. The usual structure of the Abstract is “Solved problem (1-2 phrases)” - “Solution methods” - “Obtained results”.
The abstract was accepted by journal editors.
The term "durability test" is not very well used in this context. Correct to use the term "cutting test"
The term „durability test“ has been changed to „Tool-life test“ according to ISO standard.
Since a scientific article differs from a textbook, it makes no sense to use images that do not contain new information (for example, images of standard equipment or standard processes). From this point of view, it is worth checking out Figures 1, 2 and 5. Do these figures contain anything new and useful for a professional?
We consider to present the methods of cutting edge preparation (settings, clamping, etc.).
Also, I see no point in Figure 3 (especially since this information is duplicated in more useful form in Table 5.)
It has been changed.
A review of the current state is clearly insufficient. Total examined 11 articles, of which 6 - article by one author (certainly distinguished). The language of the article is not high enough. A lot of mistakes, terminological inaccuracies, English needs significant improvement.
English has been improved by native speaker. The current state has been extended.
The result of the Introduction should be a clearly defined work goal. The goal, formulated as "the largest possible control of process parameters, a high repeatability, productivity and automation" is too general and not specific.
This phase is not our aim. The aim has been formulated in the introduction. The results of this article can be used as improvement for „process control, repeatability, productivity and automation“.
Table 3 and 4, Figure 6 - scale bars should be added.
It has been changed.
What coatings for cutting tools were used? Why were they chosen?
We used PVD coating. It is added in the article.
The authors talk about "lower residual stress" - but there are no measurements of these parameters.
The current state of the art mentions about the residual stress and its effect on the thermal cracks. The measurement of residual stress was not the object of article.
And what is a "Sample ABC"? Where are they described?
The samples are described in chapter 3 - 3. Measurement and evaluation of surface quality of cutting inserts.
Table 3 should be better structured.
It has been changed.
The article is completely raw and requires radical processing. The level of research is insufficient (better study of the properties of the surface layer after various processing methods using, in particular, cross sections is necessary, at least).
It has been improved.
The reviewer cannot recommend this article for publication.
Thank you for your comments.
Round 2
Reviewer 1 Report
The authors correctly responded to all my suggestions.
Author Response
Thank you for your review.
Best regards.
Author team.
Reviewer 2 Report
The reviewers appreciate the answers for authors but bot all the comments were considered and, besides, most of the time the authors come with some explanation as"was improved, changed, and so one " but they never indicate what was changed or why the used some notation.
Further comments are below:s
“From Abstract and introduction is not undesirable your contribution
It could be considered
The state of art is very brief and requires improvements
It has been improved.”
I have understand they were slightly improved but still is difficult to understand the author contribution from abstract and introduction.
The authors suggest that they improved the language style but for me still is very low and it need major improvements.
Besides, from reviewer knowledge the “also. Also, “ is not used in technical English. Other difficult phrase were found as “Parameters of abrasive jet machining (Figure 1) were used for the surface treatment of coated cutting inserts used for experimental activity, see Table. 1”. Please use a technical English correction editor, the “MDPI” has one that can be very helpful.
The table 5 was revised but the authors do not mention why the magnification x2000 and x5000 was used ?
Besides why the authors put the Figure in a Tables and other as just Figures…I suggest the use only Figures for the one from Tables 5 and other such kind of data (from Table 4 true colour information, the Figure from Table 6)..and put caption for every Figure as Fig (a) (b) and so on.
I appreciate the authors put a novel section of discussion and results, but how are different their results compare to literature ? Please justify it against literature.
The conclusions are much better but still poor presentation. I do not understand the link between the bullet points and above statements.
Therefore, I suggest major revision for further consideration
Author Response
The reviewers appreciate the answers for authors but bot all the comments were considered and, besides, most of the time the authors come with some explanation as"was improved, changed, and so one " but they never indicate what was changed or why the used some notation.
Further comments are below:
“From Abstract and introduction is not undesirable your contribution. The state of art is very brief and requires improvements.
The state of art and introduction have been extended to more publications and literatures deal with the same or similar topics.
The abstract was approved by editors of Coating Journal and it contains the issue of the whole article. The introduction characterizes only just current state of knowledge of the issue.
I have understand they were slightly improved but still is difficult to understand the author contribution from abstract and introduction.
The abstract summarizes issue of the article, it contains individual points which are solved during experimental activity.
The authors suggest that they improved the language style but for me still is very low and it need major improvements.
Proofreading was made by a native speaker working at Technical University with specialization in Mechanical Engineering.
Besides, from reviewer knowledge the “also. Also, “ is not used in technical English. Other difficult phrases were found as “Parameters of abrasive jet machining (Figure 1) were used for the surface treatment of coated cutting inserts used for the experimental activity, see Table. 1”. Please use a technical English correction editor, the “MDPI” has one that can be very helpful.
Thank you for a recommendation.
The table 5 was revised but the authors do not mention why the magnification x2000 and x5000 was used ?
The device with this magnification was available. Visible unevenness which has arisen after surface adjustment were visible with 2 000x magnification. The droplets were identified on the PVD surface with 5 000x magnification. Used magnification is sufficient for the both cases.
Besides why the authors put the Figure in a Tables and other as just Figures…I suggest the use only Figures for the one from Tables 5 and other such kind of data (from Table 4 true colour information, the Figure from Table 6)..and put caption for every Figure as Fig (a) (b) and so on.
This location enables better clear arrangement and description of figures than individual figures separately. The description in tables saves the space.
I appreciate the authors put a novel section of discussion and results, but how are different their results compare to literature ? Please justify it against literature.
The results are in accordance with study literatures and the conclusion of the authors are shown in Chapter 1.
The conclusions are much better but still poor presentation. I do not understand the link between the bullet points and above statements.
The bullets were used to emphasize the conclusion reached during experiment. This form is in accordance with templates of such types of papers and their form is more transparent.
Therefore, I suggest major revision for further consideration
Thank you for your review.
Reviewer 3 Report
The authors made some changes and improved the quality of the article. However, the reviewer is still unable to recommend this article for publication, as it is too superficially examining the wearout reasons and their reasons. In particular, the following notes:
«before and mainly after the coating process» - That is, the processing occurs after coating. Thus, it is likely that the TiN layer is removed and a harder and more wear-resistant TiAlN layer remains. What is the point of such processing?
The effect of such treatment on the properties of the coating cannot be investigated without a serious study of the microstructure properties of the coating BEFORE and AFTER such treatment. That is, the authors must first examine the structure of the samples with the coating before the processing (to make a cross-section and examine using SEM and, preferably, TEM). It is also necessary to repeat these tests for specimens subjected to treatment (type B and C).
The data in table 5 is not enough to compare. You can only see that the micro droplets have been removed and craters have remained in their place. What happened to the microstructure of samples B and C? How has she changed? How deep are the changes visible? Some nanometers? Whole coating thickness?
Table 3: Why was chosen such a coating: Ti-TiAlN/TiN-TiN? Why such layers?
Micrographs show poor coverage structure. No layers are visible, multi-layer structure is not visible. Need a larger image scale.
"Thermal cracks" - how did the authors understand that these cracks are associated with thermal effects?
Table 7 is also not enough. Need cross-sections, studies with a magnification of 10,000 - 20,000 and more. It is necessary to study the nature of cracking, destruction of coatings, etc.
Based on table 7 it is difficult to make any conclusions at all. Need static - 5 - 10 - 20 tests and average results. A one-off picture may be associated with random factors.
Author Response
The authors made some changes and improved the quality of the article. However, the reviewer is still unable to recommend this article for publication, as it is too superficially examining the wearout reasons and their reasons. In particular, the following notes:
«before and mainly after the coating process» - That is, the processing occurs after coating. Thus, it is likely that the TiN layer is removed and a harder and more wear-resistant TiAlN layer remains. What is the point of such processing?
No coating layer was removed during the use of cutting edge preparations. The coatings has the same color before and after preparations. The top layer TiN (gold color) was retained.
The effect of such treatment on the properties of the coating cannot be investigated without a serious study of the microstructure properties of the coating BEFORE and AFTER such treatment. That is, the authors must first examine the structure of the samples with the coating before the processing (to make a cross-section and examine using SEM and, preferably, TEM). It is also necessary to repeat these tests for specimens subjected to treatment (type B and C).
The aim of experiment was not find out of microstructure of surface coatings but the influence of cutting edge preparations on surface integrity, tool life of cutting tools and industry repeatability.
The data in table 5 is not enough to compare. You can only see that the micro droplets have been removed and craters have remained in their place. What happened to the microstructure of samples B and C? How has she changed? How deep are the changes visible? Some nanometers? Whole coating thickness?
The size of droplets is much smaller that thickness of coatings, the coatings was not removed.
Table 3: Why was chosen such a coating: Ti-TiAlN/TiN-TiN? Why such layers?
It is the most used coatings for cutting inserts suitable for milling technology.
Micrographs show poor coverage structure. No layers are visible, multi-layer structure is not visible. Need a larger image scale.
Picture was added in table 3. Approx. thickness of coated layers were mentioned.
"Thermal cracks" - how did the authors understand that these cracks are associated with thermal effects?
The thermal cracks are typical display of cutting tool wear during milling process and interrupted cut where the temperature changes occur. The force cracks could not be so much big under given milling conditions that they could be exposed.
Table 7 is also not enough. Need cross-sections, studies with a magnification of 10,000 - 20,000 and more. It is necessary to study the nature of cracking, destruction of coatings, etc.
The character of cracks and their initiation was not the object of experimental activity. The presence of cracks was used for evaluation of the influence of used cutting edge preparation on wear resistance.
Based on table 7 it is difficult to make any conclusions at all. Need static - 5 - 10 - 20 tests and average results. A one-off picture may be associated with random factors.
The number of tests 24 was sufficient for a statistical evaluation. From the point of view of the achieved results there was no large deviations. Thank you for your review.
Round 3
Reviewer 2 Report
The comments were partially addressed.
Some issue still are related to the English style.
In terms of technical is no any issue.
Reviewer 3 Report
The reviewer maintains the opinion that the absence of metallographic studies, including the study of transverse sections, reduces the value of the results.
However, in general, after the corrections and additions made by the authors, as well as based on the explanations given by the authors, the reviewer believes that the article can be recommended for publication.